



# Extending the applicability of P3D for structure determination of small molecules

Alain Ibáñez de Opakua[1] and Markus Zweckstetter[1,2]

[1]German Center for Neurodegenerative Diseases (DZNE), Von-Siebold-Str. 3a, 37075 Göttingen, Germany.

[2]Department for NMR-based Structural Biology, Max Planck Institute for Biophysical Chemistry, Am Faßberg 11, 37077 Göttingen, Germany.

*Correspondence to*: Alain Ibáñez de Opakua (Alain.Ibanez-de-Opakua@dzne.de) and Markus Zweckstetter (Markus.Zweckstetter@dzne.de)

**Abstract.** The application of anisotropic NMR parameters for the correct structural assignment of small molecules requires the use of partially ordered media. Previously we demonstrated that the use of P3D simulations using PBLG as alignment medium allows the determination of the correct diastereomer from extremely sparse NMR data. Through the analysis of the structural characteristics of small molecules in different alignment media we here show that when steric or electrostatic factors dominate the alignment, P3D-PBLG retains its diastereomer discrimination power. We also demonstrate that P3D simulations can define the different conformations of a flexible small molecule from sparse NMR data.

## 1 Introduction

Complete structure determination of small molecules, including stereochemistry, is a challenging task and an important step in organic chemistry and drug discovery. In the last years anisotropic NMR parameters, especially residual dipolar couplings (RDCs), have been used to determine the conformation of small organic molecules in organic solvents (Li et al., 2018). RDCs provide a spatial view of the relative orientations of bonds irrespective of internuclear distances being a good complement for conventional NMR restraints such as NOE distances (Anet and Bourn, 1965) and dihedral angles from 3J spin-spin coupling constants (Haasnoot et al., 1980). However, RDCs cannot be measured in isotropic conditions because of the averaging of anisotropic NMR parameters by uniform molecular tumbling (Luy and Kessler, 2006). Thus, access to anisotropic NMR parameters requires the generation of anisotropic environments in solution and this is achieved by using an alignment medium (Tjandra and Bax, 1997).

There are two major types of alignment media with different inherent mechanism to orient tumbling molecules: lyotropic liquid crystalline (LLC) phases and strain induced alignment in a gel (SAG) (Bax, 2003; Böttcher and Thiele, 2012; Canales et al., 2012; Schmidts, 2017). The LLC phases spontaneously align in the presence of strong external magnetic fields because of their large magnetic susceptibility anisotropy. This alignment is then partially transmitted to the solvent and the molecules in solution with the degree and characteristics of the alignment depending on the concentration, temperature and other parameters



of the sample (Krupp and Reggelin, 2012). To generate a LLC phase a minimal concentration is necessary and therefore a minimal alignment is introduced, which limits the tunability of the alignment strength. The degree of orientation in these media is therefore sometimes too large, complicating the extraction of RDCs. The strain induced alignment in a gel (SAG) method generates the anisotropy mechanically, by compressing or stretching the gels. The alignment is then independent of the

magnetic field and scalable over a wide range, making it easier to tune the alignment strength. However, sample preparation can be more difficult in the case of compressed and stretched gels. In addition, it often takes several days for the solute molecule to properly diffuse into the gel (Li et al., 2018).

Each alignment medium has its own degree of induced order and orientation of alignment. These properties can be influenced

by experimental conditions and the properties of the small molecule to be aligned. How this happens is largely unknown. To fill this gap, we recently developed a three-dimensional molecular alignment model termed P3D (Ibáñez de Opakua et al., 2020). P3D allows the establishment of a quantitative correlation between the atomic structure of the alignment medium, the molecular structure of the small molecule and molecule-specific anisotropic NMR parameters. For the implementation of the model we selected poly(γ-benzyl-L-glutamate) (PBLG) as alignment medium. PBLG forms a LLC phase and has a well-

defined α-helical structure (Marx and Thiele, 2009). The P3D simulation uses a combination of steric obstruction and continuum electrostatics. Analysis of several small molecules demonstrated that the P3D model reliably discriminates between different relative configurations of small molecules dissolved in PBLG (Ibáñez de Opakua et al., 2020). To gain insight into the applicability of P3D-based enantiomer discrimination when other alignment media are used, we here investigate how different the alignment of solutes dissolved in other alignment media are and how these differences are tuned by the type of

solute. A better understanding of these questions will allow the application of molecular alignment simulations performed for one alignment media to other media if the alignment is expected to be similar. In addition, an informed choice of the alignment medium might become possible when different alignments are required to discriminate between candidate structures (Ramirez and Bax, 1998).

The discrimination between different structures of a molecule is a major application of anisotropic NMR parameters (Zweckstetter and Bax, 2000). In the case of small molecules, the determination of the constitution by NMR spectroscopy is often a standard procedure, while the determination of its conformation and configuration can be more challenging. Relative configuration determination has been improved in the last years (Li et al., 2018; Liu et al., 2018; Ibáñez de Opakua et al., 2020), in particular through the use of RDCs and residual chemical shift anisotropies (RCSAs). In addition, a number of

different chiral alignment media, which possess enantiodifferentiating properties, were developed (Marx et al., 2009; Arnold et al., 2010; Krupp and Reggelin, 2012; Meyer et al., 2012; Hansmann et al., 2016; Reller et al., 2017). However, it is so far not possible to reliably predict the enantiodifferentiating properties from the molecular properties of the chiral alignment medium and the small molecule, leaving the determination of the absolute configuration on the basis of anisotropic NMR parameters an unsolved problem (Berger et al., 2012). In addition, flexible small molecules with multiple conformations



complicate the use of least-square methods to fit the experimental RDC/RCSA values to an ensemble of structures, especially if the conformers align with different alignment tensors. Here molecular alignment simulations might help to identify those structures that fit better to the experimental RDCs. We therefore also investigate in the current study, if P3D simulations are not only beneficial for the analysis of the relative configuration of small molecules, but also for their conformational analysis.

## 2 Methods

Strychnine and IPC structures were built as described previously (Strychnine: Bifulco et al., 2013; IPC: Ibáñez de Opakua et al., 2020). RDCs of the molecules in the different alignment media were obtained from the references of Table 1. Sucrose conformer structures and RDCs were obtained from (Ndukwe et al., 2019; supplemental information).

Molecular alignment simulations (P3D), implemented in the software PALES (Zweckstetter, 2008), were performed as
described previously (Ibáñez de Opakua et al., 2020). During the simulation the solute molecule is moved in steps on a three-dimensional grid that covers the central part of PBLG. At each step, an uniform distribution of different solute molecule orientations is sampled. The spacing of the three-dimensional grid was set to 0.4 Å and the number of sampled orientations to 1,800 (100 orientations on the unit sphere and 18 in the third dimension). The interaction energy between the solute molecule and the PBLG particle is then calculated for each orientation/grid position on the basis of the precomputed potential file of the
PBLG particle and the charges of the solute molecule. The interaction energy is converted into a Boltzmann weighing factor and the RDCs are calculated. Charges of the small molecules were calculated using AtomicChargeCalculator server (Ionescu et al., 2015) via the electronegativity equalization method based on a common charge calculation scheme (atoms-in-molecules) and a robust quantum mechanical approach (HF/6-311G).

The alignment tensors for strychnine and (-)-IPC in the different alignment media and for the different conformers of sucrose were calculated by best-fitting experimental RDCs to the respective structures using singular value decomposition (SVD) as implemented in the software PALES (Zweckstetter, 2008). Variations in the SVD-derived quality measures R and RQ were evaluated using a Monte Carlo noise method (Zweckstetter and Bax, 2002), in which random noise was added to the experimental RDCs according to their estimated accuracy.

In order to calculate the relative populations of each sucrose conformer from the P3D-predicted RDCs, RQ values were maximized by a grid search over the conformer populations using steps of 1 %.



### 3 Evaluation of the partial alignment of P3D-PBLG and different alignment media with two different solutes

Several different alignment media for organic solvents are nowadays available. In order to evaluate the alignment properties of these media, we selected the two most widely studied molecules in the field: strychnine and isopinocampheol (IPC). Strychnine is a reference compound for relative configuration determination because of the high number of chiral centers (six chiral centers generating 13 diastereomers). In addition, the low flexibility of strychnine minimizes contributions from different conformations. IPC is also a rigid molecule, with little overlap in the two-dimensional proton-carbon correlation spectrum and

with both enantiomers available. Because of these favorable properties, the groups of Reggelin and Thiele used IPC for the development of alignment media with enantiodifferentiation capabilities. As we are not focusing here on the absolute configuration problem, we selected the enantiomer (-)-IPC for the analysis. We further examined the literature and extracted a set of nine alignment media for each of the compounds from which five are shared. Table 1 shows the list of the 12 selected alignment media together with the references from where the data were taken. The structures of the basic units of the alignment

media are displayed in Figure 1.

**Table 1: Selected alignment media for which experimental RDCs of strychnine and (-)-IPC were reported.**

| | Strychnine | |
|---|---|---|
| PBLG | poly(γ-benzyl-L-glutamate) | Liu et al., 2018 |
| PELG | poly(γ-ethyl-L-glutamate) | Thiele, 2004 |
| PMMA | poly(methyl methacrylate) | Nath et al., 2015 |
| PIAF | poly(L-isocyanoalanyl-L-phenylalanine benzyl ester) | Li et al., 2017 |
| PS | Cross-linked polystyrene | Luy et al., 2004 |
| PA1 | Poly-A-1/L-alanine derived polyacetylene | Dama and Berger, 2012b |
| PL1 | Poly-L-1/poly(phenylisocyanide) | Dama and Berger, 2012a |
| PALV | L-valine derived polyacetylene | Meyer et al., 2012 |
| PADV | D-valine derived polyacetylene | Meyer et al., 2012 |
| | (-)-IPC (isopinocampheol) | |
| PBLG | poly(γ-benzyl-L-glutamate) | Marx et al., 2009 |
| PBDG | poly(γ-benzyl-D-glutamate) | Marx et al., 2009 |
| PELG | poly(γ-ethyl-L-glutamate) | Hansmann et al., 2016 |
| PALF300 | L-phenylalanine derived polyacetylene at 300 K | Krupp and Reggelin, 2012 |
| PALF316 | L-phenylalanine derived polyacetylene at 316 K | Krupp and Reggelin, 2012 |
| PL1 | Poly-L-1/poly(phenylisocyanide) | Reller et al., 2017 |
| PALV | L-valine derived polyacetylene | Meyer et al., 2012 |
| PADV | D-valine derived polyacetylene | Meyer et al., 2012 |
| PPEMG | poly(N-methyl-N'-((R)-1-phenylethyl)guanidine) | Arnold et al., 2010 |



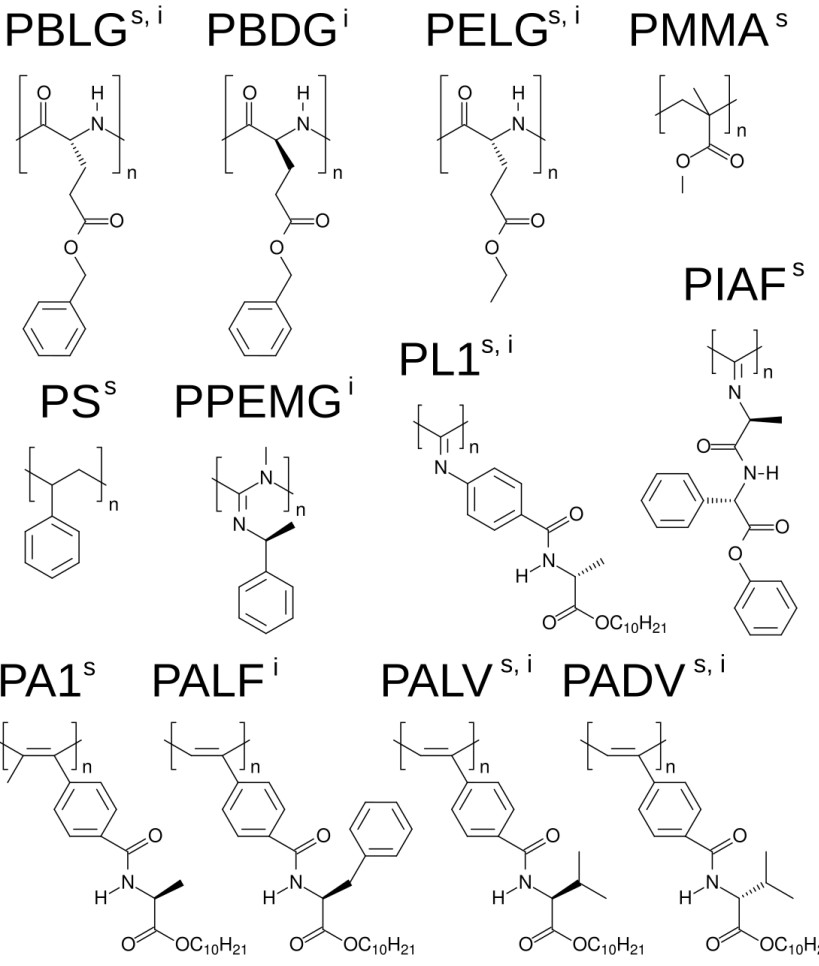

**Figure 1: Representation of the basic units of the anisotropic media used for the alignment of strychnine (s) and (-)-IPC (i).**


From the 12 selected alignment media, ten form LLC phases. PMMA and PS were used as compressed and stretched gels, respectively. The bias towards LLC phases likely arises, because a major objective of the current development of new alignment media is on enantiodifferentiation where helical chiral nonracemic polymers, capable of forming LLC phases, are good candidates. The LLC-forming polymers include polyglutamates, polyisocyanates, polyacetylenes, polyisocyanides and

polyguanidines.

The experimental one-bond CH RDCs observed for strychnine and (-)-IPC in these alignment media were compared both among each other and with the RDCs calculated by the P3D simulation using the PBLG model (Figs. 2 and 3). In the case of strychnine, half of the alignment media besides PBLG have a good correlation with the RDCs simulated by P3D on the basis



of the PBLG alignment model (Fig. 3a). These alignment media are PELG, PMMA, PIAF and PS. The experimental RDCs in the second group of alignment media (PA1, PL1, PALV, PADV) also largely correlate among each other, but deviate more from the P3D-calculated RDCs (Fig. 3a). This is confirmed by inspection of the alignment tensors (Fig. 3b): the orientation of the z axis of the P3D-calculated alignment tensor is similar to those derived by singular-value decomposition (SVD) from the experimental RDCs in many of the alignment media, with the exception of PA1 and PL1. Notably, a smaller amount of RDCs

were reported for PA1 and PL1 (five and six RDCs, respectively), which makes SVD-derived tensor orientations sensitive to the exact CH bond orientations in the employed structural models (Zweckstetter and Bax, 2002). We also point out that in some cases alignment tensor axes were swapped, e.g. the y-axis is positioned where in other alignment media the z axis is found (Fig. 3b). This can occur when two consecutives axes/eigenvalues have similar magnitude such that inaccuracies in experimental RDCs or molecular alignment simulation result in an exchange/relabeling of these axes, which however has only

little influence on the back-calculated RDCs.

### Strychnine

$R_{(7)}S_{(8)}S_{(12)}R_{(13)}R_{(14)}S_{(16)}$

| | RDCs (Hz) | | | | | | | | | |
|---|---|---|---|---|---|---|---|---|---|---|
| | P3D PBLG | PBLG | PELG | PMMA | PIAF | PS | PA1 | PL1 | PALV | PADV |
| CH1 | -114.9 | -178.9 | -43.5 | 37.0 | 19.6 | -9.3 | 8.5 | 35.0 | 27.9 | 19.9 |
| CH2 | -118.1 | -36.9 | -15.3 | 22.8 | -3.3 | 0.8 | | -0.5 | -14.9 | -11.7 |
| CH3 | -185.5 | -69.8 | -12.8 | 15.7 | 5.6 | | -16.6 | -18.4 | -11.8 | -5.2 |
| CH4 | -115.7 | -182.4 | -43.5 | 37.8 | 21.2 | -10.5 | 10.6 | 39.5 | 28.0 | 21.5 |
| CH8 | 84.5 | 87.1 | 24.6 | -17.5 | -11.0 | 3.6 | | -5.7 | | |
| CH12 | 200.1 | 146.8 | 32.4 | -40.0 | -10.0 | 11.4 | | | -4.1 | -2.6 |
| CH13 | 15.6 | 48.9 | 12.2 | -11.0 | -5.7 | -3.6 | 3.6 | | -5.8 | -4.8 |
| CH14 | 184.1 | 151.7 | 31.5 | -38.7 | -11.5 | 12.0 | | | -8.4 | -5.4 |
| CH16 | -101.1 | -49.6 | -14.8 | 25.1 | -1.9 | 1.8 | | -6.4 | | |
| CH22 | -34.1 | -1.1 | 3.0 | -2.8 | 2.5 | -6.6 | 6.0 | | 1.0 | 0.0 |

### (-)-IPC

$R_{(1)}R_{(2)}S_{(3)}$

| | RDCs (Hz) | | | | | | | | | |
|---|---|---|---|---|---|---|---|---|---|---|
| | P3D PBLG | PBLG | PBDG | PELG | PALF300 | PALF316 | PL1 | PALV | PADV | PPEMG |
| CH1 | 17.8 | 13.3 | 14.5 | 9.2 | 6.9 | 2.7 | 9.8 | -23.6 | 7.4 | 9.0 |
| CH2 | 2.9 | -7.4 | -5.2 | -12.1 | 5.0 | 4.7 | 34.2 | 13.8 | 1.0 | 12.0 |
| CH3 | 3.7 | 11.1 | 13.1 | 2.9 | 6.0 | -2.7 | 21.6 | 11.6 | 5.3 | 64.0 |
| CH5 | 0.7 | -5.5 | -5.3 | -7.3 | -10.2 | 0.1 | -25.2 | -10.3 | -14.0 | 52.0 |
| $CH_2 4$ | 1.1 | -4.0 | -4.4 | -3.4 | -7.6 | -1.3 | -19.7 | 4.6 | -2.9 | -13.0 |
| | 10.8 | 10.9 | 14.3 | -2.4 | 19.0 | 5.5 | 56.2 | -3.8 | 10.0 | 40.0 |
| $CH_2 7$ | -27.5 | -16.4 | -17.4 | -9.8 | -4.8 | 2.2 | -0.5 | 15.1 | -1.9 | |
| | -1.5 | 4.1 | 1.9 | 13.6 | 1.2 | -5.5 | -4.7 | 9.5 | 6.9 | |

**Figure 2: Strychnine and (-)-IPC structures together with the CH carbon labels and the correct configuration (left), as well as the respective lists of RDCs in the different alignment media (right).**



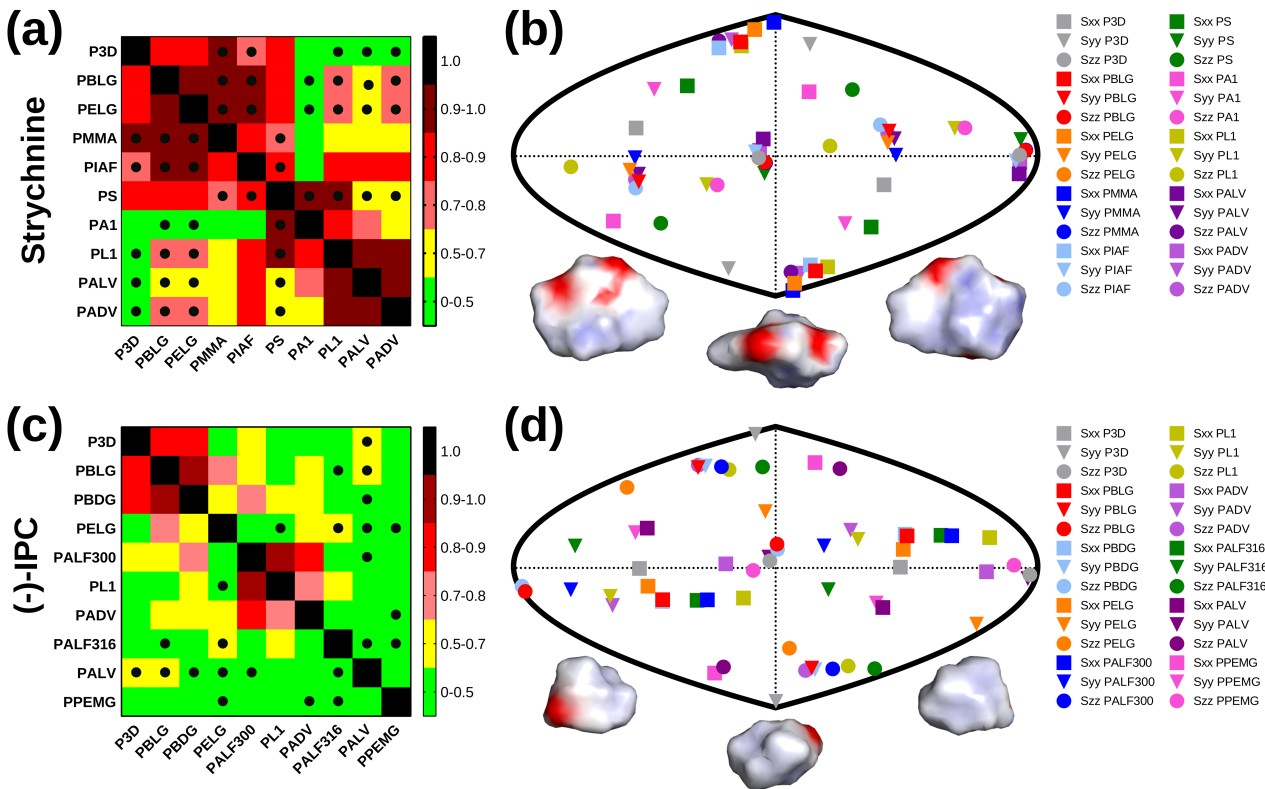

**Figure 3:** RDCs and alignment comparison of different alignment media and P3D for strychnine and (-)-IPC. (a, c) Matrices of
Pearson's correlation R between P3D-calculated RDCs and the experimental RDCs in different alignment media for strychnine (a)
and (-)-IPC (c). Dots mark negative correlations. (b, d) Comparison of the orientation of the P3D-predicted alignment tensor (grey)
with alignment tensors derived by SVD from experimental RDCs of the different alignment media for strychnine (b) and (-)-IPC
(d). The orientation of the three axes corresponding to the eigenvalues Szz (circle), Syy (triangle) and Sxx (square) of the diagonalized
alignment tensor are projected onto a two-dimensional world map. Different orientations of the charged surface of strychnine (b)
and (-)-IPC (d) are shown below.

The alignment media that correlate better with the P3D prediction were further analyzed in Figure 4. Comparison of P3D-
predicted and experimental RDCs (Fig. 4a), as well as the alignment tensors projected onto a two-dimensional world map (Fig.
4c) and the aligned strychnine structures (Fig. 4d), show that the partial ordering of strychnine in these alignment media are
very similar and well predicted by P3D. Experimental RDCs observed in PMMA have a very good correlation with the P3D-
calculated RDCs (Fig. 3a, 4a) and also a very similar alignment tensor (Fig. 4c), but the correlation is negative (Fig 3a, dots).
The negative slope indicates that the major alignment axis in PMMA is oriented orthogonal to the field, while PBLG aligns

**MAGNETIC RESONANCE**
Discussions
with its helix axis parallel to the magnetic field. Indeed, the PMMA gel was compressed, while the stretched PS gel displayed a positive correlation with the P3D-calculated RDCs. In other words, strychnine has in PMMA a highly similar alignment tensor as in PBLG/PS but with an opposite sign of the axial component of the alignment ($D_a$). For these reasons, when different alignment media are being compared, we use here the absolute value of the Pearson's correlation coefficient R.

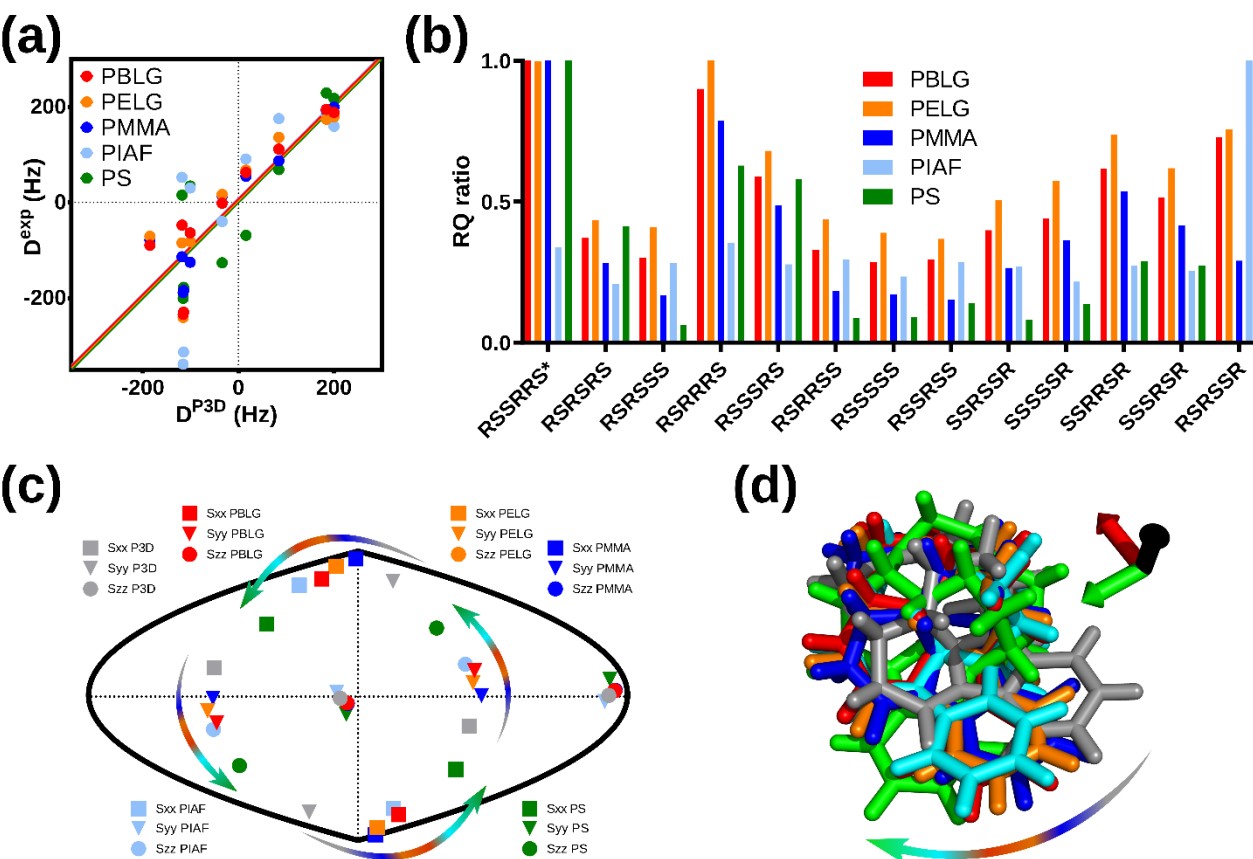

**Figure 4: P3D and the alignment media with most similar alignment properties for strychnine. (a) Correlation between P3D-simulated RDCs ($D^{P3D}$) and experimental RDCs ($D^{exp}$) with different alignment media for strychnine. Experimental values are normalized by the slope of the linear fitting. (b) Diastereomer discrimination power of different alignment media for strychnine based on the P3D simulation and using the RQ ratio for judging the quality of correlation. (c) Comparison of the orientation of the P3D-predicted alignment tensor (grey) of strychnine with alignment tensors derived by SVD from the experimental RDCs using PALES (Zweckstetter, 2008) observed in different alignment media. The orientation of the three axes corresponding to the eigenvalues Szz (circle), Syy (triangle) and Sxx (square) of the diagonalized alignment tensor are projected onto a two-dimensional world map. (d) Oriented structures of strychnine according to the PBLG-based P3D simulation and the different experimentally analyzed alignment media. Axes colors are black (z), red (x) and green (y), and are rotated when the axes of the alignment tensor**



**are swapped because of a similar magnitude of the corresponding eigenvalues. The arrows in (c) and (d) illustrate the change in the orientation of the x and y axes in different alignment media.**


We then investigated the diastereomer discrimination power of P3D-PBLG when using experimental RDCs observed in strychnine dissolved in the five different alignment media (Fig. 4b). To enhance the discrimination power we use the quality parameter RQ, defined as $(R+1)^2/Q_S$, where $Q_S$ is the RDC quality factor $Q = rms(D^{exp}-D^{P3D})/rms(D^{exp})$ scaled by the slope of the $D^{exp}$ *vs* $D^{P3D}$ fitting (Ibáñez de Opakua et al., 2020). The ability of P3D-PBLG to select the correct diastereomer is retained

for alignment of strychnine in PELG, PMMA and PS. This is not the case for PIAF, which has a smaller Pearson's R (0.71) when compared to PELG/PMMA/PS (all over 0.8) (Fig. 3a). This suggests that R values larger than 0.8 are needed to identify the correct diastereomer, in agreement with the previously published P3D-based diastereomer discrimination analysis for six different small molecules (Ibáñez de Opakua et al., 2020).

Next, we performed the same P3D-based analysis for (-)-IPC, which has different alignment properties when compared to strychnine. The correlation matrix for (-)-IPC only shows a strong correlation of P3D with PBLG and PBDG (Fig. 3c). In agreement with a weaker enantiodiscrimination power of PBLG when compared to PELG (Hansmann et al., 2016), or the other helical chiral nonracemic polymers shown here, the experimental RDCs observed for (-)-IPC in PBLG and PELG differ. On the other hand, PBLG and PBDG induce similar alignment such that the discrimination of different diastereomers of (-)-

IPC was retained for PBDG. Notably, a change in the enantiomer of the alignment medium (e. g. from PBLG to PBDG) has the same effect as changing the enantiomer of the solute (e.g. from (-)-IPC to (+)-IPC) (Marx et al., 2009).

As far as correlations between experimental RDCs in different alignment media are concerned, only few media induce similar alignment of (-)-IPC (Fig. 3c). Only PL1, PALF300 and PADV form a small cluster in the correlation matrix. A correlation

between experimental RDCs induced by PADV and PL1 was present for both (-)-IPC and strychnine with R values of 0.71 and 0.92, respectively. The pronounced differences in the alignment of (-)-IPC in different alignment media is also evident from the comparison of the respective alignment tensors: the projected axes orientations do not cluster in certain regions (Fig. 3d), in contrast to the alignment tensors of strychnine (Fig. 3b). Further notable are the RDC differences when (-)-IPC is aligned in PALF300 and PALF316 (R=0.45). This L-phenylalanine derived polyacetylene forms different LLC phases at

different temperatures, with the helical structure severely disrupted at 316 K (Krupp and Reggelin, 2012). The pronounced differences in the alignment of (-)-IPC in PALF300 and PALF316 indicates that for certain molecules/alignment media fine structural details of the alignment medium are critical for enantiodiscrimination.

In order to rationalize the distinct alignment properties of strychnine and IPC, we analyzed the structural properties of the two

molecules (Fig. 3b,d). While strychnine has an oval disc-like shape, the shape of IPC is quite spherical, and both molecules have asymmetric charge distributions (Fig. 3b,d). Comparison of the correlation coefficients of the experimental RDCs with



RDCs predicted by molecular alignment simulation using P3D or only steric interactions (1D obstruction model; Zweckstetter and Bax, 2000) suggested that electrostatic interactions are more important for the alignment of strychnine: R values dropped from 0.88 to 0.64 in the case of strychnine and from 0.84 to 0.68 in the case of (-)-IPC, when replacing P3D simulations by 1D obstruction model simulations.

The excellent correlation between P3D-predicted and experimental RDCs of strychnine in PBLG indicates that the alignment of strychnine in PBLG is dominated by steric and electrostatic factors (Fig. 3, 4). At the same time, the fine structural details of the alignment media appear to be less important, which results in similar alignment of strychnine in PBLG, PELG, PMMA and PS (Fig. 4). On the other hand, the quite spherical shape of IPC suggests that steric obstruction is less important for its molecular alignment. Instead specific molecular interactions between IPC and the alignment medium become relevant and are responsible for the differences in RDC values observed in different alignment media. The difference in the alignment of IPC in PBLG and PELG might be correlated with the stronger enantiodifferentiating power of PELG, which has been linked to the change in the bulkiness and mobility of the lateral side chain (Hansmann et al., 2016). Due to these differences, IPC can have more and stronger diastereomorphous interactions with the chiral helical backbone of PELG. The importance of fine structural details for the alignment of IPC also provides a rationale for why IPC is an excellent test molecule to study the enantiodifferentiation properties of alignment media.

## 4 Analysis of conformational ensembles using P3D

Because strychnine and IPC are rigid molecules, a single alignment tensor accurately describes their weak LLC/gel-induced alignment. However, for more flexible molecules it is necessary to determine alignment tensors for all the conformers or independently for every flexible part of the molecule (Thiele and Berger, 2003). In order to simplify this problem, linearly independent alignment media would be needed (Ramirez and Bax, 1998), which, as shown in Fig. 3, is not always easy to achieve. When only one alignment medium is available, selection of energetically more promising structures and back-calculation of anisotropic NMR parameters to best-fit experimental values might be used (Tzvetkova et al., 2019). The latter approach, however, requires a large amount of anisotropic NMR parameters and becomes difficult when the alignment tensors of the conformers are different.

We previously developed the P3D alignment simulation to solve the relative configuration problem, demonstrating that P3D can identify the correct diastereoisomer from a very small number of RDCs, even with less than five RDCs, the minimum number of RDCs required for SVD. Here we now use P3D to address the problem of conformation. To this end, we selected sucrose (Fig. 5a), which has recently been analyzed by anisotropic NMR in PBLG (Ndukwe et al., 2019). Studies based on molecular dynamics (MD) simulations and solution NMR suggested the presence of multiple sucrose conformations (Delaglio et al., 2005; Xia and Case, 2012). On the basis of 11 RDCs and 12 RCSAs, the conformational ensemble of sucrose in



CDCl₃/DMSO (70:30) was best described with three conformers, which were selected from a set of low energy DFT structures
(Ndukwe et al., 2019). The respective ΔG of conformers 1, 2 and 3 were 0, 2.04 and 2.72 kcal/mol, with conformer 3 being highly similar to the crystal structure of sucrose (Russo et al., 2013).

Instead of the 23 anisotropic NMR parameters used by Ndukwe and colleagues, we use here only the eight one-bond CH RDCs (Fig. 5a,b). Following the same rationale as before (Ibáñez de Opakua et al., 2020), we selected the one-bond CH RDCs (Fig.
5b) because they are the largest RDCs in small molecules, i.e. can be measured with high accuracy, and there is less ambiguity in the assignment. We then subjected the three conformers of sucrose to P3D alignment simulation. P3D-simulated RDCs were averaged over the three-member ensemble and compared with the experimental RDCs (Fig. 5b,c). The results indicate that the three-member ensemble of conformers improved the correlation reaching a R value of 0.975 (Fig. 5c). The RQ of the weighted average is also significantly larger than the RQ values for any of the three individual conformers (Fig. 5d).




Open Access MAGNETIC RESONANCE Discussions

**(a)**

**(b)**

|  | RDCs (Hz) | | | | | | | |
|---|---|---|---|---|---|---|---|---|
|  | CH1 | CH2 | CH3 | CH4 | CH5 | CH9 | CH10 | CH11 |
| Conformer 1 (33%) | 43.22 | -16.83 | -13.40 | -11.59 | 34.53 | 36.78 | 34.34 | 34.65 |
| Conformer 2 (35%) | 42.23 | -8.60 | -25.31 | 52.39 | 53.20 | 51.00 | 49.61 | 48.10 |
| Conformer 3 (32%) | 59.48 | -19.65 | -27.44 | -10.79 | -0.53 | 0.60 | -1.66 | 1.27 |
| Weighted average | 48.08 | -14.85 | -22.06 | 11.06 | 29.85 | 30.18 | 28.17 | 28.68 |
| Experimental | 52.25 | -17.74 | -19.79 | 3.12 | 22.23 | 23.32 | 21.3 | 17.53 |

**(c)**

**(d)**

**(e)**

**Figure 5: Validation of the conformational ensemble of sucrose using P3D. (a) Sucrose structure with the CH carbons labeled. (b) List of P3D-simulated RDCs for the 3 conformers together with the average weighted RDCs and the experimental RDCs. (c) Correlations between the experimental RDCs ($D^{exp}$) and the P3D-simulated RDCs for the 3 conformers and the weighted average. (d) RQ ratios of the 3 different conformers in reference to the weighted average. Error bars are calculated as the propagation of R and $Q_S$ errors and these errors are calculated from the std of 100 repetitions including noise in the RDCs. (e) Comparison of the**




**orientation of the P3D-predicted alignment tensors for the 3 conformers. The orientation of the three axes corresponding to the eigenvalues Szz (circle), Syy (triangle) and Sxx (square) of the diagonalized alignment tensor are projected onto a two-dimensional**
**world map.**

The structures of the three conformers were aligned before the simulation in order to have the same molecular frame and be able to compare the alignment tensors (Fig. 5e). The result shows that all the conformers have a similar but not identical alignment, indicating that the differences in RDCs come mainly from the structural differences, in agreement with the use of
the variable-weight single-tensor SVD method to solve the conformational structure of the molecule (Ndukwe et al., 2019).

While the variable-weight single-tensor SVD method strongly relies on the assumption that different conformers have similar alignment tensors, this is not required for the P3D-based conformational analysis. We therefore determined the relative amounts of conformers by maximizing the P3D-based RQ parameter (Fig. 6a,b). The comparison of the RQ ratios of the RQ
maximized 3-conformer ensemble and the RQ maximized 2-conformer ensembles (Fig. 6c) shows that three is the minimum number of conformers to get an almost perfect fit (R=0.996; $Q_S$=0.076). In addition, the contribution of conformer 3 was increased to 49 % (32% in (Ndukwe et al., 2019)) in the refined 3-conformer ensemble (Fig. 6a,b). Notably, the most populated conformer (conformer 3) is closest to the crystal structure of sucrose.






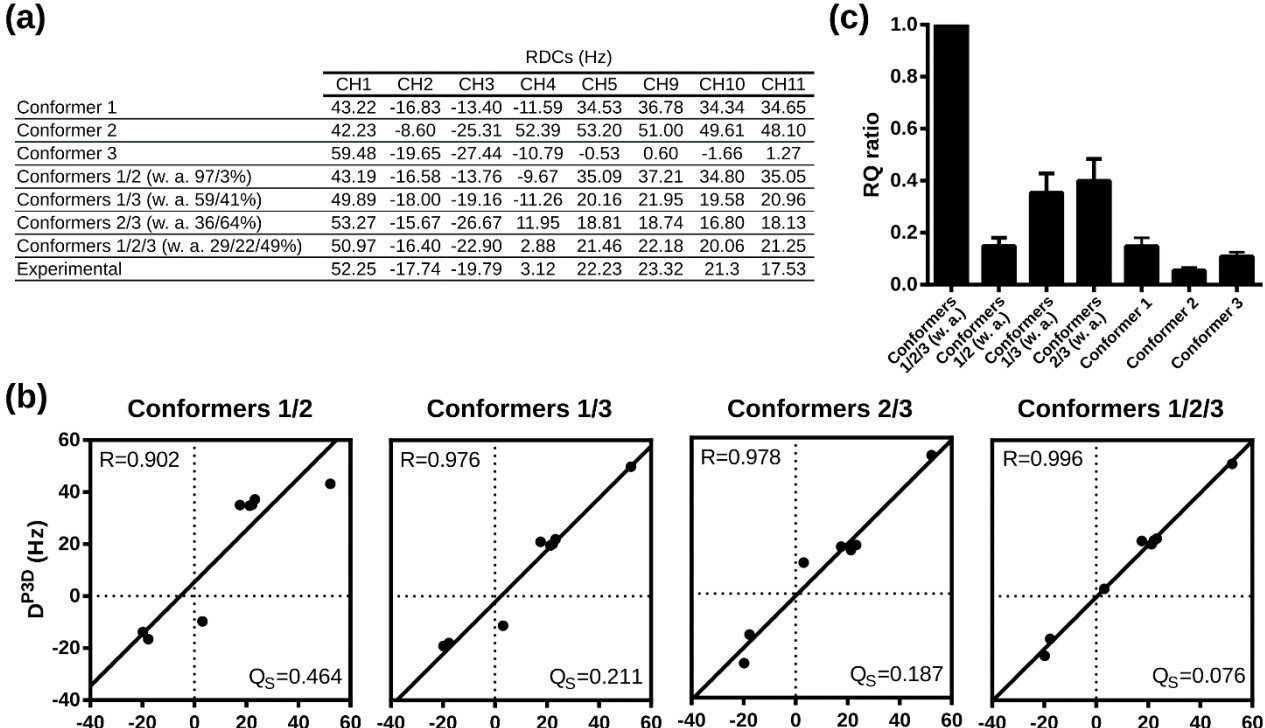

**Figure 6: P3D-based refinement of the conformational ensemble of sucrose. (a) List of P3D-calculated RDCs for the 3 conformers together with the average weighted RDCs from the RQ maximization for 2- and 3-conformer ensembles and the experimental RDCs. (b) Correlation between the experimental RDCs ($D^{exp}$) and the P3D-calculated RDCs for the weighted average of the RQ maximization for 2- and 3-conformer ensembles. (c) RQ ratios of the three different conformers in reference to their weighted average as 2- and 3-conformer ensembles. Error bars were calculated as the propagation of R and $Q_S$ errors and these errors are calculated from the std of 100 repetitions including noise in the RDCs.**

## 5 Conclusions

The current study highlights the important role of molecular alignment simulations for the structural analysis of small molecules. In agreement with previous data (Ibáñez de Opakua et al., 2020), the new analysis supports the applicability of P3D simulations for the determination of the relative configurations, but also extends it to the analysis of conformational ensembles of small molecules. In addition, molecular alignment simulations might – with further improvements – become crucial for the determination of the absolute configuration. While much progress has been made in the development of powerful chiral alignment media, this is restricted to the differentiation between enantiomers, similar to exposing small molecules to polarized light. To determine the absolute configuration, atomistic descriptions are required that link the NMR anisotropic parameters obtained from chiral alignment media with the correct enantiomer. A next step towards this goal could be the inclusion of



specific interactions between the solute and the alignment medium, for example salt bridges, into molecular alignment simulations.

Towards this next step, it is important to define an alignment medium, which has a structure amenable to structural modeling and strong enantiodiscrimination capabilities. We therefore analyzed different alignment media and compared them with our P3D alignment simulation model of PBLG. Literature search identified only two molecules, strychnine and IPC, for which RDCs in several different alignment media had been reported. The results of our analysis suggested that the weak alignment induced by LLC phases and gels critically depends on both the molecular properties of the alignment medium and the small

molecule. The comparison further showed that the alignment of IPC varies more strongly across the available alignment media when compared to strychnine (Fig. 3). We interpret this as a consequence of the more symmetrical/spherical shape of IPC such that more specific interactions with the alignment medium more strongly contribute to the alignment process. In the case of strychnine, on the other hand, steric obstruction together with electrostatic interactions dominate molecular ordering. This leads to less variability in the partial alignment of the small molecule allowing the application of the P3D-PBLG model to

other alignment media (Fig. 4). The choice of alignment medium should therefore take into account the structural properties of the small molecule of interest, especially its shape as well as the charge distribution.

We also investigated the applicability of the P3D-PBLG simulation approach to the challenge of determining conformational ensembles of flexible small molecules. On the example of sucrose, we showed that P3D can be used to determine the

populations of different conformers in an ensemble, with the advantage that it can work even when individual conformers have different alignment tensors. Through the P3D-based analysis we optimized the population of each conformer in the ensemble. The analysis resulted in an ensemble in which the population of conformer 3, which is closest to the crystal structure of sucrose, was increased to almost 50 % (Fig. 6). The lower population of conformer 3 in the SVD-based ensemble might arise from the slightly different alignment tensors of the three conformers (Fig. 5e).


In summary, P3D alignment simulations establish a quantitative connection between the alignment medium, the molecular structure of small molecules and anisotropy-based NMR parameters. P3D can therefore predict RDCs for different alignment media depending on the structural details of both the alignment medium and the small molecule and thus enables the determination of conformational ensembles of flexible small molecules.


**Data availability.** All data that support the findings of this study are available from the corresponding authors upon reasonable request.

**Author contribution.** AIdeO and MZ designed the project and wrote the paper. AIdeO conducted the data analysis.






**Competing interests.** The authors declare that they have no conflict of interest.

**Financial support.** M.Z. was supported by the Helmholtz society through the Impuls- und Vernetzungsfond 'Multiscale Bioimaging: From Molecular Machines to Networks of Excitable Cells', and by the advanced grant '787679 - LLPS-NMR'
of the European Research Council.

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
