# Peer review of "Extending the applicability of P3D for structure determination of small molecules"

_Magnetic Resonance, 2020_

## Referee Comment (RC1) · Anonymous Referee #1 · 19 Dec 2020

The paper submitted by M. Zweckstetter applies molecular alignment simulations (P3D) to the structure determination of small molecules by comparing predicted residual dipolar couplings (RDCs) with experimental data.

Before I will comment further on the paper I have to say that this manuscript is close to being unreadable without an intense recourse to the first paper on the subject published by the same author in Angew. Chemie. Furthermore, the manuscript does not include any Supporting Information, which additionally compromises the comprehension of its scientific content. This means that this manuscript cannot stand on its own which is a very unpleasant fact for the reader (and the reviewer!). Moreover, the introduction

should be rewritten in order to give credit to the people doing pioneering work in the field. Ad Bax may have coined the term RDCs, but this doesn't mean that he invented the technique or that the paper of Bax and Tjandra was a seminal one. Courtieu, Prestegard, Lesot and of course Emsley to name just a few, should be mentioned here. [1-4] In general the citation policy of the manuscript is far from being of good scientific practice (the LLC-phase literature, [5-7] the structure of PBLG – its helicity is known since 1954[8] not 2009 and so on...). But apart from these formal deficiencies, the scientific value of the paper is more than questionable.

P3D simulations are based on MD derived PBLG snapshot geometries, around which a cubic grid is placed. The molecular geometries of the analytes under investigation (strychnine, IPC, and sucrose) are placed on this grid, and Boltzmann-averaged RDC sampling is carried out by evaluating the averages over all grid points and all analyte orientations. The interaction between the analytes and the alignment polymer PBLG are evaluated purely on the basis of static (pre-computed grids) interactions that include steric (excluded volume) and electrostatic (based atomic charges) terms. This type of simulation represents a fast, but surely very crude model to simulate the analyte-polymer interactions as well as the alignment process. Not only are vdW-interactions completely ignored, but also all dynamic and entropic contributions to the alignment process are neglected. The rather coarse-grained grid used in the simulations (grid spacing 0.4A) and the rough sampling of molecular orientations (1800 per grid point) must necessarily lead to large uncertainties. It is not even clear, whether a cubic grid superimposed to a rod-shaped, cylindrical polymer may introduce systematic errors. Certainly, the excluded-volume simulations are apt to introduce large degrees of order even at large polymer-analyte distances when first contacts become possible. The Boltzmann-averaging is highly sensitive towards energies used, and simple electrostatic interactions using static molecular models are with some certainty crude oversimplifications. The RDCs obtained from the P3D-PBLG simulation are then compared to experimental data obtained from diverse alignment media (Figure 2), though obviously these media do have vastly differing alignment properties (see Figure 2, RDCs

across different alignment media also differ vastly in their magnitude). It is unclear how the P3D derived RDCs were scaled to account for different degrees of order, and how a PBLG simulation should compare to chemically different alignment media such as PELG, PMMA, PS, PA, etc. Even for the simulated and experimental data of PBLG, there are huge discrepancies between the individual RDCs of up to 115 Hz (Figure 2, strychnine, CH3 RDC)!!! The invalid comparisons are continued in Figure 3, were matrices of Pearson correlations are given in color-coded form. Many of these correlations are negative (marked by dots in Figure 3), and thus raise additional doubts on the assumptions made by the P3D simulations (by the way: the color-code used in Figure 3 is also highly misleading, as "green" color obviously indicate bad correlations). The obviously invalid cross-alignment media / P3D – PBLG comparisons are then continued in Figure 4 for different diastereomers of strychnine, yet it remains unclear how such a crude alignment simulation that neglects almost all relevant interactions (including all dynamic alignment polymer properties) can differentiate the molecular configurations. The investigations are then extended to the more flexible structure of sucrose, were only eight out of 23 RDCs that have been reported in the literature have been used. It is open to speculation why only this small subset of experimental data is used – may be the rest of the data doesn't fit well? Three different sucrose conformers are evaluated, the geometries of which were taken from the literature. Figure 5 details the results for the three individual sucrose conformers, where large deviations of the experimental and calculated RDCs are observed indicated by significant deviations of the correlations from the diagonal of the plots given in Figure 5c. The relative contributions of the sucrose conformers are then optimized by maximizing the RQ parameter, and an "an almost perfect fit (R=0.996; QS=0.076)" was finally obtained (Figure 6). However, given the sparsity of the NMR data used, and the number of conformers evaluated, it is clear that a multi-conformer fit represents sort of an over-fitting scenario, which is not supported by an adequate amount of experimental data. Given the wealth of conformational data available for a common compound such as sucrose, a more thorough evaluation against the literature data available is mandatory. The most highly populated structure of sucrose seems to be close to its solid-state conformation, but this must not necessarily be the correct description for the flexibility of sucrose as the solution conformation may differ significantly therefrom. The claim stated in the conclusion of this paper that "molecular alignment simulations might – with further improvements – become crucial for the determination of the absolute configuration" is, based on these results, an unjustified expectation as these simulations supposedly must treat molecular interactions on a much finer and much more detailed level, which even may be out-of-reach altogether at least in the (near) future. In view of the roughness of the model, the complete neglect of dynamic effects, the over-simplifications of the molecular interactions, and the invalidity of the cross-alignment comparisons employed here I cannot see how these P3D simulations may be used to elucidate even the relative configuration of slightly more complex natural products of unknown configuration beyond reasonable doubt. For these reasons publication in Magnetic Resonance is not recommended.

[1] I. Canet, J. Courtieu, A. Loewenstein, A. Meddour, J. M. Pechine, "Enantiomeric analysis in a polypeptide lyotropic liquid crystal by deuterium NMR", J. Am. Chem. Soc. 1995, 117, 6520-6526. [2] P. Lesot, Y. Gounelle, D. Merlet, A. Loewenstein, J. Courtieu, "Measurement and Analysis of the Molecular Ordering Tensors of 2 Enantiomers Oriented in a Polypeptide Liquid-Crystalline System", J. Phys. Chem. 1995, 99, 14871-14875. [3] P. Lesot, D. Merlet, A. Meddour, J. Courtieu, A. Loewenstein, "Visualization of Enantiomers in a Polypeptide Liquid-Crystal Solvent through C-13 Nmr-Spectroscopy", Journal of the Chemical Society-Faraday Transactions 1995, 91, 1371-1375. [4] P. Lesot, D. Merlet, J. Courtieu, J. W. Emsley, "Discrimination and analysis of the NMR spectra of enantiomers dissolved in chiral liquid crystal solvents through 2D correlation experiments", Liq. Cryst. 1996, 21, 427-435. [5] A. Krupp, M. Noll, M. Reggelin, "Valine derived poly (acetylenes) as versatile chiral lyotropic liquid crystalline alignment media for RDC-based structure elucidations", Magn. Reson. Chem. 2020, 1-10. DOI: 10.1002/mrc.5003. [6] P. Lesot, P. Berdague, A. Meddour, A. Kreiter, M. Noll, M. Reggelin, "H-2 and C-13 NMR-Based Enantiodetection Using

Polyacetylene versus Polypeptide Aligning Media: Versatile and Complementary Tools for Chemists", Chempluschem 2019, 84, 144-153. [7] M. Leyendecker, N. C. Meyer, C. M. Thiele, "Development of New Supramolecular Lyotropic Liquid Crystals and Their Application as Alignment Media for Organic Compounds", Angew. Chem. Int. Ed. 2017, 56, 11471-11474. DOI: 10.1002/anie.201705642, # 40829. [8] P. Doty, A. M. Holtzer, J. H. Bradbury, E. R. Blout, "Polypeptides .2. The Configuration of Polymers of Gamma-Benzyl-L-Glutamate in Solution", J. Am. Chem. Soc. 1954, 76, 4493-4494. DOI 10.1021/ja01646a079.

––––––––––––––––––––––––––––––

---

## Referee Comment (RC2) · Anonymous Referee #2 · 21 Dec 2020

This manuscript reports extensive testing of simulations of RDC data by a software program, P3D, particularly as applied to data collected in lyotropic liquid crystal alignment media used in diastereomeric characterization of small molecules. Much effort is being expended in the development of new alignment media for small molecules, often with the goal of enantiomeric identification. The later is not achievable when only RDC and RCSA data are used. Prospects are much better if molecular characteristic of solute – alignment interactions can be included. This is what the P3D software attempts to do. The extensive testing presented here, on several solutes and several recently introduced alignment media takes a step toward realizing this goal. Some useful observations, include the fact that predictions work well when solutes depart from a near

spherical structure or when electrostatic interactions are strong and asymmetric. Also, the program appears to be useful in modeling systems having multiple conformations having individual alignment tensors. There are, however, some additional issues that could be discussed. While potential reasons for failure associated with solutes of high symmetry are well discussed, less attention is given to reasons for failure of about half the media tested on a well-behaved solute. Also, there are presumably factors associated with the nature of the RDC sets (number of measurements and degeneracy of dipole orientations). Some comments regarding these could be added. There are some discrepancies in the sucrose data that should be clarified. In figure 5 the numbering of the sucrose structure (a) and the notations in the table (b) don't agree. CH vectors with numbers 1,2,3 and 4 in the structure displayed are nearly parallel and should have very similar RDCs. The data appear to come from the Ndukwe reference, which does show this trend. In the manuscript table RDCs with the Ndukwe values are numbered 5,9, 10 and 11. Also, it is not clear where the populations are coming from. These appear to be fitted parameters? The text seems to suggest that the populations are consistent with free ender estimates on line 315. They are not. Also, some comment might be made in comparison to water MD simulations (Case) where only M1 and M2 are highly populated with a difference of only 0.3kcal, something more in line with populations. There are a few places that the text could be improved for clarity: Line 31 – not clearly worded. Maybe: "alignment requires a minimum concentration of lyotropic medium and then often aligns strongly at this concentration, resulting in .." line 87: RQ might be defined here as opposed to much later.

---

## Author Comment (AC1) · 6 Jan 2021

1st reviewer

Before I will comment further on the paper I have to say that this manuscript is close to being unreadable without an intense recourse to the first paper on the subject published by the same author in Angew. Chemie.

Reply: As stated in the manuscript, the focus of the current work is on "extending the applicability of the P3D simulation model". The P3D simulation has already been published in Angew Chem and key features of the method were summarized in the

current manuscript. We are convinced that this procedure is fully in agreement with scientific practice.

Furthermore, the manuscript does not include any Supporting Information, which additionally compromises the comprehension of its scientific content.

Reply: We included all results in the tables and figures of the manuscript.

This means that this manuscript cannot stand on its own which is a very unpleasant fact for the reader (and the reviewer!).

Reply: Scientific findings/publications are generally based on previous studies and the corresponding information is provided through references. In addition, we summarized the key aspects of the P3D simulation, which was previously published in Angew Chem, in the section "Methods" of the manuscript.

Moreover, the introduction should be rewritten in order to give credit to the people doing pioneering work in the field. Ad Bax may have coined the term RDCs, but this doesn't mean that he invented the technique or that the paper of Bax and Tjandra was a seminal one. Courtieu, Prestegard, Lesot and of course Emsley to name just a few, should be mentioned here. [1-4] In general the citation policy of the manuscript is far from being of good scientific practice (the LLC-phase literature, [5-7] the structure of PBLG – its helicity is known since 1954[8] not 2009 and so on. . .). But apart from these formal deficiencies, the scientific value of the paper is more than questionable.

Reply: We added the suggested references.

P3D simulations are based on MD derived PBLG snapshot geometries, around which a cubic grid is placed. The molecular geometries of the analytes under investigation (strychnine, IPC, and sucrose) are placed on this grid, and Boltzmann-averaged RDC sampling is carried out by evaluating the averages over all grid points and all analyte orientations. The interaction between the analytes and the alignment polymer PBLG are evaluated purely on the basis of static (pre-computed grids) interactions

that include steric (excluded volume) and electrostatic (based atomic charges) terms. This type of simulation represents a fast, but surely very crude model to simulate the analyte-polymer interactions as well as the alignment process. Not only are vdW interactions completely ignored, but also all dynamic and entropic contributions to the alignment process are neglected. The rather coarse-grained grid used in the simulations (grid spacing 0.4A) and the rough sampling of molecular orientations (1800 per grid point) must necessarily lead to large uncertainties. It is not even clear, whether a cubic grid superimposed to a rod-shaped, cylindrical polymer may introduce systematic errors. Certainly, the excluded-volume simulations are apt to introduce large degrees of order even at large polymer-analyte distances when first contacts become possible. The Boltzmann-averaging is highly sensitive towards energies used, and simple electrostatic interactions using static molecular models are with some certainty crude oversimplifications.

Reply: In fact, in our previous publication, one of the most exciting conclusions was that just the steric and electrostatic factors are able to discriminate the correct diastereomer and we showed that with 6 different small molecules. The grid spacing and the number of molecular orientations were selected as described in our Angew Chem paper, i.e. smaller grid spacings or increased numbers of orientations did not significantly change the predicted RDCs.

The RDCs obtained from the P3D-PBLG simulation are then compared to experimental data obtained from diverse alignment media (Figure 2), though obviously these media do have vastly differing alignment properties (see Figure 2, RDCs across different alignment media also differ vastly in their magnitude). It is unclear how the P3D derived RDCs were scaled to account for different degrees of order

Reply: Please note that the different magnitude is irrelevant when the Pearson correlation coefficient (R) is used. In addition, when calculating the Q factor, the RDCs were normalized by the slope of the linear fitting (as stated on line 90: "the RDC quality factor Q = rms(Dexp-DP3D)/rms(Dexp) scaled by the slope of the Dexp vs DP3D fitting")

And how a PBLG simulation should compare to chemically different alignment media such as PELG, PMMA, PS, PA, etc.

Reply: One of the aims of the study is to analyze how the small molecules align in different alignment media, that's why we compare the alignment in chemically different alignment media using the simulated PBLG as reference.

Even for the simulated and experimental data of PBLG, there are huge discrepancies between the individual RDCs of up to 115 Hz (Figure 2, strychnine, CH3 RDC)!!!

Reply: This data is the same as in our previous publication. Of course, errors in the simulation can affect more some of the RDCs than others. This is a consequence of the fact that we didn't optimize the parameters independently for each of the simulations in order to avoid a bias. While we do not claim that our simulations are perfect, the correlation between experimental and P3D-predicted RDCs are consistently of high quality (as demonstrated by high Pearson's correlation coefficients).

The invalid comparisons are continued in Figure 3, were matrices of Pearson correlations are given in color-coded form. Many of these correlations are negative (marked by dots in Figure 3), and thus raise additional doubts on the assumptions made by the P3D simulations

Reply: We described in the manuscript the reason of negative correlations for different alignment media (line 236): "The negative slope indicates that the major alignment axis in PMMA is oriented orthogonal to the field, while PBLG aligns with its helix axis parallel to the magnetic field. Indeed, the PMMA gel was compressed, while the stretched PS gel displayed a positive correlation with the P3D-calculated RDCs. In other words, strychnine has in PMMA a highly similar alignment tensor as in PBLG/PS but with an opposite sign of the axial component of the alignment (Da)."

By the way: the color-code used in Figure 3 is also highly misleading, as "green" color obviously indicate bad correlations.

Reply: We used red as hot color to indicate a good correlation. In the revised version of the manuscript, we changed green to cyan in Fig. 3.

The obviously invalid cross-alignment media/P3D – PBLG comparisons are then continued in Figure 4 for different diastereomers of strychnine, yet it remains unclear how such a crude alignment simulation that neglects almost all relevant interactions (including all dynamic alignment polymer properties) can differentiate the molecular configurations.

Reply: Please, read our Angew Chem paper, in which we demonstrate with 6 different molecules & several different tests that the P3D simulation indeed works.

The investigations are then extended to the more flexible structure of sucrose, were only eight out of 23 RDCs that have been reported in the literature have been used. It is open to speculation why only this small subset of experimental data is used – may be the rest of the data doesn't fit well?

Reply: With due respect, this is not open to speculation; we explain why in the manuscript (lines 322-324): "Following the same rationale as before (Ibáñez de Opakua et al., 2020), we selected the one-bond CH RDCs (Fig. 5b) because they are the largest RDCs in small molecules, i.e. can be measured with high accuracy, and there is less ambiguity in the assignment." Please also note that the 23 anisotropic NMR parameters, which were reported by Ndukwe and colleagues, are not only RDCs, but include 12 RCSAs. This is also stated on lines 316-318 of our manuscript: "On the basis of 11 RDCs and 12 RCSAs, the conformational ensemble of sucrose in . . . (Ndukwe et al., 2019)." From these 11 RDCs, 3 belong to averaged RDCs from both geminal protons of a $CH_2$ group that are not independently assigned. Thus, no cherry picking was done.

Three different sucrose conformers are evaluated, the geometries of which were taken from the literature. Figure 5 details the results for the three individual sucrose conformers, where large deviations of the experimental and calculated RDCs are observed indicated by significant deviations of the correlations from the diagonal of the plots given in Figure 5c. The relative contributions of the sucrose conformers are then optimized by maximizing the RQ parameter, and an "an almost perfect fit (R=0.996; QS=0.076)" was finally obtained (Figure 6). However, given the sparsity of the NMR data used, and the number of conformers evaluated, it is clear that a multi-conformer fit represents sort of an over-fitting scenario, which is not supported by an adequate amount of experimental data.

Reply: A better agreement between experimental and predicted RDCs is of course reached as one increases the degrees of freedom. But the main aim here was to reproduce - with sparse data - the results obtained using SVD (by Ndukwe and colleagues). In contrast to SVD, the P3D-based approach does not rely on the assumption that the 3 conformers have an identical alignment tensor.

Given the wealth of conformational data available for a common compound such as sucrose, a more thorough evaluation against the literature data available is mandatory. The most highly populated structure of sucrose seems to be close to its solid-state conformation, but this must not necessarily be the correct description for the flexibility of sucrose as the solution conformation may differ significantly therefrom. The claim stated in the conclusion of this paper that "molecular alignment simulations might – with further improvements – become crucial for the determination of the absolute configuration" is, based on these results, an unjustified expectation as these simulations supposedly must treat molecular interactions on a much finer and much more detailed level, which even may be out-of-reach altogether at least in the (near) future.

Reply: Thanks for the suggestion. To better stress the need for further improvements, we state in the revised version of the manuscript: "To determine the absolute configuration, atomistic descriptions are required that link the NMR anisotropic parameters obtained from chiral alignment media with the correct enantiomer. A next step towards this goal could be the inclusion of specific interactions between the solute and the alignment medium, for example salt bridges, into molecular alignment simulations."

In view of the roughness of the model, the complete neglect of dynamic effects, the over-simplifications of the molecular interactions, and the invalidity of the cross-alignment comparisons employed here I cannot see how these P3D simulations may be used to elucidate even the relative configuration of slightly more complex natural products of unknown configuration beyond reasonable doubt. For these reasons publication in Magnetic Resonance is not recommended.

Reply: With due respect, we do not agree. While certain interactions are currently not used in the P3D simulations, P3D simulations do take into account the molecular structure of the alignment medium and the solute, as well as steric and electrostatic interactions. In the Angew Chem paper we also performed several tests, which demonstrated that the simulations are robust against dynamic changes in the structure of the alignment medium and the solute. In addition, we already showed in the Angew Chem paper that the relative configuration of small molecules can be determined using P3D. Therefore this is not the focus of the current manuscript.

---

## Author Comment (AC2) · 6 Jan 2021

There are some discrepancies in the sucrose data that should be clarified. In figure 5 the numbering of the sucrose structure (a) and the notations in the table (b) don't agree. CH vectors with numbers 1,2,3 and 4 in the structure displayed are nearly parallel and should have very similar RDCs. The data appear to come from the Ndukwe reference, which does show this trend. In the manuscript table RDCs with the Ndukwe values are numbered 5,9, 10 and 11.

Reply: Thanks for detecting this mistake. We corrected the numbers in the revised version of the manuscript.

[Figure]

**MRD**

Also, it is not clear where the populations are coming from. These appear to be fitted parameters? The text seems to suggest that the populations are consistent with free ender estimates on line 315. They are not. Also, some comment might be made in comparison to water MD simulations (Case) where only M1 and M2 are highly populated with a difference of only 0.3kcal, something more in line with populations.

Reply: The populations of Fig. 5 come from the Ndukwe reference and the populations of Fig. 6 are fitted. We added a comment to the figure legend to avoid this confusion. The populations are based on the three conformations present in the M1 (called S3-i, S3-iii and S3-iv in that paper). The structures were taken from table S14. The description of the populations is just a short summary of the work from Ndukwe et al. We are not trying to suggest that the free energy from DFT explains the calculated populations. The information is just descriptive.

There are a few places that the text could be improved for clarity: Line 31 – not clearly worded. Maybe: "alignment requires a minimum concentration of lyotropic medium and then often aligns strongly at this concentration, resulting in .."

Reply: Thanks, we changed it to: "Alignment requires a minimum concentration of lyotropic medium and often aligns strongly at this concentration, which limits the tunability of the alignment strength."

line 87: RQ might be defined here as opposed to much later.

Reply: Changed.

---

## Editor Decision (ED1)

Dear Dr Zweckstetter,

Two additional experts have re-evaluated your manuscript, with opposing views (as before). We therefore asked a third reviewer.

One reviewer raised several concerns, as you can read in report #1, but considers your contribution overall acceptable, but only after revision.

The other reviewer raised similar concerns as one of the reviewers before. This reviewer questions the general applicability and robustness of the model used in the P3D calculations and would not consider it an established method yet. This reviewer also finds the current manuscript no major advance vs your previous paper in Angew. Chemie and not presenting sufficient detail to reproduce the results.

The third reviewer is positive, stressed the advance in this manuscript vs your previous paper.

Considering the reports, I conclude that after a revision your contribution can be a worthy contribution for this Special Issue for Robert Kaptein in MR. I have attached all 3 reviews.

A new manuscript needs to address the concerns and comments raised by the reviewer #1 and #3 and repeating his words I would "urge you to cast your approach as a promising approach, tested on a single molecule, and thus as a preliminary but potentially feasible method to address the problem of determining the ensemble distribution of a flexible molecule by liquid crystal NMR".
Please also consider reviewer #2, by presenting more detail to reproduce the results. Would it be possible to make the program P3D available via nmrbox.org or github, and present used parameters for the calculation in the supplement? This would also be in line with the comment by reviewer #3.

I hope that you will be able and willing to do so.

With kind regards, Rolf Boelens

**mr-2020-32 Report #1**

This manuscript describes a potentially very interesting extension to the recent introduction of the P3D method that predicts the alignment of small molecules in liquid crystals, somewhat analogous to the PALES program that has been widely used for proteins. It is perhaps surprising that P3D works as well as it does because the surface characteristic of the various alignment media is not accurately known at the atomic level, whereas this must play a role in aligning the solute (as highlighted by the brief discussion of (-)-IPC in PALF300 and PALF316). Considering that in many cases it does work fine, as recently published in Angew. Chemie, I have no problem with accepting that fact.

A significant fraction of the current manuscript focuses on a very long-standing problem: Can liquid crystal or otherwise anisotropic NMR define the ensemble distribution of flexible molecules? This question has been studied for nearly 50 years, including by some of the giants of the magnetic resonance field (incl. Luz, Pines, Jim Emsley), but with mixed success (nicely reviewed by J. Emsley in the Encyclopedia of NMR). P3D now changes the approach because it uses the shape of the conformer to directly predict its alignment, thereby removing the key problem that the number of adjustable parameters steeply increases with the number of conformers considered. The flipside, however, is that P3D predictions are quantitatively not all that precise. I also note that the agreement is provided in terms of an RQ parameter that does not account for the alignment strength, which is most definitely needed when calculating the ensemble populations. It also is not clear how sensitive the method is to the accuracy at which P3D predicts the alignment of a given conformer, and therefore, whether it is statistically warranted to select individual conformers, and their populations, from an effectively infinite ensemble.

Although I believe the current manuscript is suitable for publication in Magnetic Resonance, I strongly urge the authors to cast these efforts as a promising approach, tested on a single molecule, and therefore a preliminary but potentially feasible method to address the long-standing problem of determining the ensemble distribution of a flexible molecule by liquid crystal NMR.

A few minor points:
1. For sucrose, the authors decided to focus only on the one-bond C-H RDCs. However, I note that for the glucose ring, four of the CH-bonds are effectively collinear, reducing the number of independent observables to five.
2. The authors repeated point out that their highest energy conformer is close to the crystal structure, perhaps suggesting it is a reasonable conformation despite being 2.7 kcal higher in energy than their lowest energy conformer. Some more discussion on why the actual energies are only rough estimates may be appropriate.
3. To what extent are the sucrose conformers compatible with other parameters (e.g. JHH, NOEs, RDCs) of this extensively studied molecule (see e.g. Freedberg JACS 124, 2358, 2002 and Carb. Res. 340, 863, 2005 and references cited therein)?
4. Strain-induced alignment in a gel (SAG) was first (simultaneously) introduced by Tycko (JACS) and by Grzesiek (J Biomol NMR), and that perhaps should be clarified in the referencing.
5. Is it possible to show by cross validation that indeed the 3-member ensemble provides an improved fit (treating the 4 parallel Glc RDCs as 1)?

The manuscript by Ibáñez de Opakua and Zweckstetter describes the assignment of the relative configuration of three compounds (strychnine, IPC, and sucrose) using anisotropic NMR parameters. The anisotropic NMR parameters used in this investigation are one-bond RDCs. The RDCs (literature data) are translated into structural informational by the author's own computer program P3D. This is a continuation report of a previous publication in ACIE (acie 2020, 59, 6172-6176).

The problem for the reader is that the authors do not give enough information about their method (P3D). This was already true for their first paper in ACIE on this topic (acie 2020, 59, 6172-6176). It needs to be further mentioned that two out of the six compounds from the ACIE paper were used again (strychnine and IPC). Therefore, the title of the manuscript ("Extending the applicability of P3D for structure determination of small molecules") seems to be totally exaggerated. The three known compounds of the manuscript were already studied as model systems many times and are by no means an extension of the applicability!

The results for the two compounds from the first paper (strychnine and IPC) are given in Figure 2. The experimental RDCs in PBLG are given in column 2 and the calculated RDCs by P3D in column 1. The agreement of these values is not very good. Usually the results are presented as so-called D(calc) plots (experimental versus predicted RDCs). Furthermore, no quality factors or anything related are given. In the ACIE 2020 paper, they have given the R-values for strychnine (0.88) and IPC (0.84). It is surprising how the authors can even attempt a configurational assignment of both compounds given these large deviations for the correct configuration.

It remains unclear for the reader why the authors have added eight columns for strychnine and IPC (further alignment media). The values do not contribute anything to the manuscript (although a long discussion is included here) because you cannot compare experimental RDCs from alignment medium B with the calculated RDCs from alignment medium A. This comparison is invalid!

For the third molecule (sucrose), the authors used the RDC data of Ndukwe et al. (cc 2019, 55, 4327-4330). They applied only eight out of the 23 originally described RDCs ("one-bond CH RDCs"). Given the limited number of RDCs (in respect to the assignment of the relative configuration, the conformation, and the conformational ensembles), additional fitting parameters (molar fractions) must necessarily lead to an increase in the fitting quality between the experimental and simulated data.

In total, the P3D method is a very rough and coarse-grained model that reduces the interactions of the alignment medium with the molecules under investigation to purely static van der Waals and electrostatic interactions. All dynamic aspects of the polymer in a condensed LLC phase are discarded.

Due to the many objections mentioned above, the manuscript should be rejected.

**mr-2020-32 Report #3**

Compared to the previous paper the in Angewandte Chemie, https://doi.org/10.1002/anie.202000311 this manuscript describes two new additions to the P3D method:

1) It assesses its applicability to other alignment media.

2) It extends the method to extract conformational ensembles in case on a mixture of conformations in solution.

Related to 1), did the authors tried to parametrize other alignment media in their P3D-PALES software? I would thing that the electrostatic component might vary in some cases?

Related to 2), the authors chose to test their method using three conformations of sucrose previously shown to be representative of what is found in solution based on previous NMR experiments. But what if they were to use more conformations? Would they still converge to three and to the same ones?

In the current era of open science, I find the statement about data availability rather strange. The authors should provide all data and scripts to allow to repeat the current work, ideally in e.g. a GitHub repository. If the authors have different policies about distributing the software itself, this is fine. But the data for this work should be made public, together with the scripts to reproduce it. Only then can other easily use this new tool.

---

## Author Response (AR4)

**Answers to the reviewers**

**mr-2020-32 Report #1**

A significant fraction of the current manuscript focuses on a very long-standing problem: Can liquid crystal or otherwise anisotropic NMR define the ensemble distribution of flexible molecules? This question has been studied for nearly 50 years, including by some of the giants of the magnetic resonance field (incl. Luz, Pines, Jim Emsley), but with mixed success (nicely reviewed by J. Emsley in the Encyclopedia of NMR). P3D now changes the approach because it uses the shape of the conformer to directly predict its alignment, thereby removing the key problem that the number of adjustable parameters steeply increases with the number of conformers considered.

**Reply:** We thank the reviewer for highlighting the importance of our work.

The flipside, however, is that P3D predictions are quantitatively not all that precise. I also note that the agreement is provided in terms of an RQ parameter that does not account for the alignment strength, which is most definitely needed when calculating the ensemble populations. It also is not clear how sensitive the method is to the accuracy at which P3D predicts the alignment of a given conformer, and therefore, whether it is statistically warranted to select individual conformers, and their populations, from an effectively infinite ensemble. Although I believe the current manuscript is suitable for publication in Magnetic Resonance, I strongly urge the authors to cast these efforts as a promising approach, tested on a single molecule, and therefore a preliminary but potentially feasible method to address the longstanding problem of determining the ensemble distribution of a flexible molecule by liquid crystal NMR.

**Reply:** We modified the final conclusion in accordance with the reviewer comment: "*As tested with the example of sucrose, P3D is also a promising approach and a preliminary but potentially feasible method for the determination of conformational ensembles of flexible small molecules.*"

A few minor points: 1. For sucrose, the authors decided to focus only on the one-bond C-H RDCs. However, I note that for the glucose ring, four of the CH-bonds are effectively collinear, reducing the number of independent observables to five.

**Reply:** In order to solve the question about the reduced number of RDCs we included a sentence with the results obtained using all RDCs: "When all RDCs are considered we get R=0.926 for the average with 0.806, 0.615 and 0.838 respectively for each conformer. The obtained populations (26/20/54%) are very similar to the ones obtained with the one-bond CH RDCs only."

2. The authors repeated point out that their highest energy conformer is close to the cristal structure, perhaps suggesting it is a reasonable conformation despite being 2.7 kcal higher in energy than their lowest energy conformer. Some more discussion on why the actual energies are only rough estimates may be appropriate.

**Reply:** The free energies of the conformers come from the Ndukwe et al. reference, from DFT calculations. Even for them, the energies do not fit exactly with the obtained

populations, but can discriminate the conformers when the energy difference is big. The step of energy calculations is needed to reduce the number of conformers to be considered, excluding the ones with too big energies, but that doesn't mean that, in the experimental conditions, the calculations are exact.

3. To what extent are the sucrose conformers compatible with other parameters (e.g. JHH, NOEs, RDCs) of this extensively studied molecule (see e.g. Freedberg JACS 124, 2358, 2002 and Carb. Res. 340, 863, 2005 and references cited therein)?

**Reply:** The aim of our work is to investigate the ability of P3D to characterize multiple conformations in small molecules. For that we took a published paper with data obtained using our selected alignment media (i.e. PBLG) and compared it with our simulations. Comparing with other results (including RDCs in different conditions) is beyond the scope of the current work. However, the current analysis demonstrates that the crystal structure of sucrose fits well to the P3D predicted RDCs.

4. Strain-induced alignment in a gel (SAG) was first (simultaneously) introduced by Tycko (JACS) and by Grzesiek (J Biomol NMR), and that perhaps should be clarified in the referencing.

**Reply:** We included the suggested references, thanks.

5. Is it possible to show by cross validation that indeed the 3-member ensemble provides an improved fit (treating the 4 parallel Glc RDCs as 1)?

**Reply:** As indicated in the first point, we decided to solve this question including the results for all RDCs.

**mr-2020-32 Report #2**

The manuscript by Ibáñez de Opakua and Zweckstetter describes the assignment of the relative configuration of three compounds (strychnine, IPC, and sucrose) using anisotropic NMR parameters. The anisotropic NMR parameters used in this investigation are one-bond RDCs. The RDCs (literature data) are translated into structural informational by the author's own computer program P3D. This is a continuation report of a previous publication in ACIE (acie 2020, 59, 6172-6176). The problem for the reader is that the authors do not give enough information about their method (P3D). This was already true for their first paper in ACIE on this topic (acie 2020, 59, 6172-6176). It needs to be further mentioned that two out of the six compounds from the ACIE paper were used again (strychnine and IPC). Therefore, the title of the manuscript ("Extending the applicability of P3D for structure determination of small molecules") seems to be totally exaggerated. The three known compounds of the manuscript were already studied as model systems many times and are by no means an extension of the applicability!

**Reply:** Please note that the point is not that we used again two out of the six compounds from our ACIE paper, but that we address two new aspects. The first one is if the P3D-PBLG simulation is applicable to other alignment media. This of course can only be

tested for small molecules, for which RDCs in several different alignment media are available (i.e. strychnine and IPC). The second new aspect, which is evaluated in the current manuscript, is if it is possible to use P3D for the conformational analysis of flexible small molecules. This is an important question as stated by reviewer #1 *("A significant fraction of the current manuscript focuses on a very long-standing problem: Can liquid crystal or otherwise anisotropic NMR define the ensemble distribution of flexible molecules? This question has been studied for nearly 50 years, including by some of the giants of the magnetic resonance field (incl. Luz, Pines, Jim Emsley), but with mixed success (nicely reviewed by J. Emsley in the Encyclopedia of NMR).")*.

It remains unclear for the reader why the authors have added eight columns for strychnine and IPC (further alignment media). The values do not contribute anything to the manuscript (although a long discussion is included here) because you cannot compare experimental RDCs from alignment medium B with the calculated RDCs from alignment medium A. This comparison is invalid!

**Reply:** In fact, the comparison was one of the aims of the study. We compared the RDCs obtained from simulations with experimental RDCs from different alignment media to identify those media that have similar alignment properties as PBLG and those which don't (and thus are either amendable to P3D prediction or not).

For the third molecule (sucrose), the authors used the RDC data of Ndukwe et al. (cc 2019, 55, 4327-4330). They applied only eight out of the 23 originally described RDCs ("one-bond CH RDCs"). Given the limited number of RDCs (in respect to the assignment of the relative configuration, the conformation, and the conformational ensembles), additional fitting parameters (molar fractions) must necessarily lead to an increase in the fitting quality between the experimental and simulated data.

**Reply:** To address this issue, we included the results obtained using all RDCs: "When all RDCs are considered we get R=0.926 for the average with 0.806, 0.615 and 0.838 respectively for each conformer. The obtained populations (26/20/54%) are very similar to the ones obtained with the one-bond CH RDCs only."

In total, the P3D method is a very rough and coarse-grained model that reduces the interactions of the alignment medium with the molecules under investigation to purely static van der Waals and electrostatic interactions. All dynamic aspects of the polymer in a condensed LLC phase are discarded.

**Reply:** What is pointed out here is in fact one important conclusion of our work: the dynamic aspects and specific interactions are less important for the alignment in PBLG and the alignment of small molecules (at least those tested here and in our Angew. Chem. Paper) is dominated by ("coarse-grained") steric and electrostatic effects.

**mr-2020-32 Report #3**

Related to 1), did the authors tried to parametrize other alignment media in their P3D-PALES software? I would think that the electrostatic component might vary in some cases?

**Reply:** While this is an interesting suggestion it is currently not possible to implement. The chosen alignment medium, PBLG, has a defined alpha-helical structure, i.e. a prerequisite for structure-based alignment prediction. In order to implement other alignment media, high quality structural models would be required.

Related to 2), the authors chose to test their method using three conformations of sucrose previously shown to be representative of what is found in solution based on previous NMR experiments. But what if they were to use more conformations? Would they still converge to three and to the same ones?

**Reply:** Using too many conformations is dangerous because of the overfitting, so it is possible to get a good fitting even with wrong conformers if too many conformers are being used. That's why an additional filter, like energy calculations, should be included.

In the current era of open science, I find the statement about data availability rather strange. The authors should provide all data and scripts to allow to repeat the current work, ideally in e.g. a GitHub repository. If the authors have different policies about distributing the software itself, this is fine. But the data for this work should be made public, together with the scripts to reproduce it. Only then can other easily use this new tool.

**Reply:** Thanks for the suggestion. We added to the methods section the command used to run our simulations with a brief description of the involved files. The P3D algorithm is available as part of the PALES program, which can be downloaded from the PALES webpage
(https://www3.mpibpc.mpg.de/groups/zweckstetter/_links/software_pales.htm).